# The Effect of Whole-Body Electromyostimulation Program on Physical Performance and Selected Cardiometabolic Markers in Obese Young Females

**DOI:** 10.3390/medicina60020230

**Published:** 2024-01-29

**Authors:** Amal Salhi, Nejmeddine Ouerghi, Hassane Zouhal, Mohamed Baaziz, Amine Salhi, Fatma Zohra Ben Salah, Abderraouf Ben Abderrahman

**Affiliations:** 1Department of Medicine Physical and Functional Rehabilitation, National Institute of Orthopedics “M.T. Kassab”, Tunis 2010, Tunisia; 2Higher Institute of Sport and Physical Education of Ksar-Said, University of Manouba, Tunis 2010, Tunisia; 3High Institute of Sport and Physical Education of Kef, UR13JS01, University of Jendouba, Kef 7100, Tunisia; najm_ouerghi@hotmail.com; 4Faculty of Medicine of Tunis, Rabta Hospital, LR99ES11, University of Tunis El Manar, Tunis 1007, Tunisia; aminesalhi316@gmail.com; 5High Institute of Sport and Physical Education of Gafsa, University of Gafsa, Gafsa 2100, Tunisia; 6M2S (Laboratoire Mouvement, Sport, Santé)—EA 1274, Université Rennes, 35000 Rennes, France; hassane.zouhal@univ-rennes2.fr; 7Institut International des Sciences du Sport (2I2S), 35850 Irodouer, France; 8Tunisian Research Laboratory “Sports Performance Optimization”, National Center of Medicine and Science in Sports (CNMSS) LR09SEP01, Tunis 2010, Tunisia

**Keywords:** cardiometabolic health, obesity, physical performance, female, whole-body electromyostimulation

## Abstract

*Background and Objectives*: Whole-body electromyostimulation is under investigation as a potential aid for obesity-related health problems, supplementing a comprehensive, evidence-based obesity management strategy that includes lifestyle, diet, and exercise. The study investigated the impact of a whole-body electromyostimulation training program on physical performance and cardiometabolic markers in young obese females. *Materials and Methods*: Twenty-eight obese females, aged over 18 years with BMI over 30 and body fat over 28% and no underlying health conditions or medication, were divided into a whole-body electromyostimulation group (15 participants) and a control group (13 participants). The whole-body electromyostimulation program lasted 12 weeks, with two 20 min sessions weekly, using bipolar, rectangular current. Assessments pre and post intervention included body composition, blood pressure, lipid profile, C-reactive protein levels, maximal oxygen consumption, and jumping and sprint performance. Two-way ANOVA and *t*-tests were used for analysis. *Results*: Statistical analysis revealed significant (group × time) interactions for body composition, systolic blood pressure, maximal oxygen consumption, jumping and sprint performance, and plasma levels of lipids and C-reactive protein. Post hoc analyses for the whole-body electromyostimulation group indicated improvements in body composition indices (*p* < 0.01), systolic blood pressure (*p* = 0.003), maximal oxygen consumption (*p* = 0.010), and both jumping and 30 m sprint performance (*p* < 0.001 and *p* = 0.001, respectively) after the intervention. Furthermore, plasma levels of lipids (*p* < 0.01) and C-reactive protein (*p* = 0.002) showed significant improvements following the training program. In contrast, no significant changes were observed for these variables in the control group. *Conclusions*: A 12-week whole-body electromyostimulation program significantly improved body composition (skeletal muscle mass, body mass index, body fat, and waist circumference), physical performance (maximal oxygen consumption, jumping and sprint performance), and certain cardiometabolic (plasma level of lipids) and inflammatory markers (C-reactive protein) in obese young women. Further research is needed to explore the broader effects of whole-body electromyostimulation on physical and cardiometabolic health.

## 1. Introduction

The global prevalence of obesity has nearly doubled in recent decades, becoming a significant public health concern [1]. Over 1.9 billion persons aged 18 years and older (39% of men and 40% of women) were overweight in 2016. More than 650 million individuals of them were obese. Between 1975 and 2016, the prevalence of obesity almost tripled. Obesity is recognized as a modifiable risk factor for cardiovascular diseases and overall mortality [2]. It is also directly linked to various diseases, including pulmonary, skeletal, and neurological disorders [3].

Regular physical activity stands out as one of the most effective strategies to prevent and treat obesity, as well as to enhance physical and mental health. Different modalities of physical exercise training (whether endurance, resistance, or high-intensity intermittent training) can improve physical fitness and various cardiometabolic risk factors. These include body composition, lipid disorders, chronic inflammation, and insulin resistance in individuals with overweight or obesity [4,5,6]. However, for many obese patients, initiating physical activity programs can be challenging due to respiratory, metabolic, cardiovascular, muscular, or joint issues that affect aerobic capacity and effort tolerance [7,8]. The rise of technological conveniences that promote sedentary behaviors, along with personal constraints like time shortages, further hinder physical activity plans [9].

Whole-body electromyostimulation (WB-EMS) offers a potential training alternative to these barriers [8]. Particularly suitable for those with limited time and low propensity for physical exercise, WB-EMS is a time-saving and highly adaptable training device [10,11]. It facilitates simultaneous contractions across major muscle groups by applying an electrical current during specialized exercises [7]. Recent studies have showcased its positive effects on physical fitness parameters in both trained and untrained individuals, improving factors like vertical jump performance and maximal oxygen consumption (VO_2max_) [12]. Recent research suggests WB-EMS can be as effective as high-intensity resistance training in promoting muscular hypertrophy, strength, and fat reduction in athletes [13]. In fact, WB-EMS training improved strength and power in young healthy individuals [8,14]. Notably, most of these studies involved healthy individuals, either sedentary or athletes [7,8,13,14]. While limited data exist on WB-EMS’s effectiveness in obese populations, available studies indicate benefits in body composition, strength, and specific metabolic indices (such as an increase in HDL-cholesterol level), particularly in obese adults with moderate to high cardiometabolic risk [8,14,15,16,17]. The available evidence suggests that WB-EMS can be a valuable tool in the battle against obesity and related health concerns [8,14,15,16,17]. However, other observations negated the possibility of this improvement, with Park et al. proving no significant metabolic adaptations [18]. Research on specific effects of WB-EMS among young individuals with obesity is still an evolving area. The available literature may vary in terms of study design, participant characteristics, and measured outcomes.

Despite the well-established efficacy of local electrical muscle stimulation (EMS) in improving physical fitness and health, there exists a significant lack of literature, especially concerning whole-body EMS (WB-EMS), particularly in the context of obese women.

Therefore, further studies are needed to confirm the role of this training method as a supplemental method to support and enhance traditional training programs in managing obesity and improving life quality for obese subjects. It could serve as a foundational step, helping deconditioned individuals gradually build strength, endurance, and overall fitness.

The present study seeks to assess the impacts of a 12-week WB-EMS training regimen on physical performance metrics and selected cardiometabolic indices in obese young females. We posit that the WB-EMS training program will improve specific health markers, reducing levels of lipids and C-reactive protein, while enhancing physical performance, adiposity, and blood pressure in this demographic.

## 2. Materials and Methods

### 2.1. Trial Design and Participants

The study’s experimental procedure followed a two-arm randomized trial with parallel groups, and there were no protocol modifications from the study’s commencement. Reporting adhered to the Consolidated Standards of Reporting Trials (CONSORT) guidelines, specifying standard items for interventional trials. The study was conducted from 15 September 2020 to 30 December 2020, during which temperatures ranged between 12 °C and 26 °C and humidity levels varied from 72% to 81%. Notably, data concerning cardiometabolic markers and physical fitness were collected after 12 weeks of WB-EMS training. The study protocol was fully approved by the local ethics committee of the Tunisian National Center for Sports Medicine and Sciences (approval number LR09SEP01, 14 September 2020).

We recruited 31 obese young females for the study. These participants were randomly divided in two groups: an WB-EMS group (n = 16) and a control group (CG, n = 15). Out of the initially selected 31 participants, three females decided to withdraw from the study (one from the WB-EMS group and two from the control group). Ultimately, 28 young females (average age: 18.71 ± 0.90 years; height: 1.61 ± 0.06 m; body mass index: 35.6 ± 2.11 kg/m^2^; body fat percentage (%BF): 31.5 ± 2.77%) completed the intervention. Of these, 15 were in the WB-EMS group (Table 1) and 13 were in the control group (Table 1) (Figure 1). The WB-EMS group underwent the WB-EMS intervention twice a week, while the control group performed the same exercises as the WB-EMS group but without using electrical stimulation.

The inclusion criteria for participant eligibility encompassed women aged 18 years and older. Prospective participants were required to have no existing medical issues or be taking medications that could significantly impact body fat or skeletal muscle mass. A BMI higher than 30 and body fat percentage exceeding 28%, indicative of obesity, were other criteria for inclusion. Additionally, participants needed to have regular menstrual cycles without abnormalities. Finally, individuals were expected to commit to not being absent for more than 2 weeks during the research period [8].

All young females were instructed to maintain their regular eating habits throughout the study. This research adhered to the guidelines of the Helsinki Declaration. Written informed consent was obtained from all participants. Throughout the study, participants were encouraged to adhere to the prescribed training regimen, attend scheduled assessments, and communicate any challenges faced during the intervention. Strict adherence to the study protocol ensured the accuracy and reliability of the data collected, contributing to the robustness of the findings. No adverse effects or specific injuries were reported in relation to the WB-EMS intervention.

Before beginning the experimental process, all subjects were familiarized with the testing tools. Anthropometric parameters, physical performance measures, systolic (SBP) and diastolic (DBP) blood pressures, as well as biochemical markers (including plasma total cholesterol, high-density lipoprotein cholesterol (HDL-C), low-density lipoprotein cholesterol (LDL-C), triglycerides, and C-reactive protein (CRP)) were assessed for both the WB-EMS and control groups at baseline and again after the 12-week period (the time between the last session and final assessment was 4 days). The same investigators carried out all of the measurements during the experimental protocol. Additionally, independent investigators were blinded in performing the statistical analyses.

### 2.2. Intervention

The experiment was overseen by licensed sports therapists with expertise in WB-EMS intervention. Each session commenced with a 5 min warm-up on a bike ergometer at moderate intensity. The WB-EMS program, set at 85 Hz with a duration of 350 μs, was operated intermittently, offering a 6 s impulse phase followed by a 4 s rest. The waveform was rectangular and delivered in a bipolar fashion. This program was designed to target eight muscle groups: the upper arms, chest, abdomen, latissimus, upper back, lower back, buttocks, and thighs. The stimulation was applied to the skin in proximity to the dermal tissue using wireless electrodes via Bluetooth technology [19]. Participants engaged in two 20 min sessions per week for 12 weeks (24 sessions in total with at least 2 days’ rest between sessions). In each session, they performed basic exercises like squats, executing 2 sets with 8 repetitions each (Table 2), and the exercises followed the same order as presented below. The intensity of WB-EMS was gauged using the Borg CR 10 scale [20], aiming for a range between 5 and 7, which was described as “hard” to “very hard”. The intensity was checked every 3 min, adjusting as needed to increase the stimulus and counter the body’s natural adaptation [21]. Each participant trained on the same days and at the same times during the intervention. The training sessions were private for each woman.

The control group performed the same exercises as the WB-EMS group but without using electrical stimulation.

### 2.3. Mesurements

#### 2.3.1. Anthropometric Assessment

Participants’ body height was measured using an unstretched tape measure while they stood without shoes. Weight-related metrics, including body mass, BMI, body fat percentage (%BF), and skeletal muscle mass (SMM), were determined using a diagnostic scale (Tanita-DC-430U) [22]. During the test, individuals stood on the Tanita scale without shoes and wearing light clothes, and the device measured the impedance of the electrical signal as it travelled through the body. This information was then used to estimate the proportions of different body components. Waist circumference (WC) was measured using a non-deformable tape ruler, placed between the superior iliac crest and the lower rib margin [23].

#### 2.3.2. Blood Pressure Assessment

Subjects were seated with their backs supported and feet flat on the floor. After a 15 min rest, measurements were taken on the left arm. Blood pressure, both systolic (SBP) and diastolic (DBP), was measured using an arm tensiometer (Exacto KD 591; Biosynex, Strasbourg, France).

#### 2.3.3. Physical Measures

Incremental running test

The assessment of maximal oxygen consumption (VO_2max_) and maximal aerobic speed (MAS) was conducted through the Vameval test on a 400 m running track, delineated by 20 cones. The test’s initiation involved participants running at a steady speed of 8 km/h. Subsequently, every minute, participants increased their running velocity by 0.5 km/h until reaching the point of fatigue. To synchronize with the predetermined pace, participants adjusted their running speed upon hearing a specific audio signal, ensuring alignment with a corresponding cone on the track. The test’s termination occurred when a participant was unable to sustain the required pace for two consecutive audio signals, signifying fatigue or an inability to maintain the specified intensity [24]. VO_2max_ was estimated based on the participant’s performance during the test. The relationship between oxygen consumption and exercise intensity was used to extrapolate the maximum oxygen uptake. Throughout the test, a heart rate monitor (S810; Polar, Kempele, Finland) was used to track the heart rate of the participants.

Performance Tests

All participants were required to complete three attempts for each test. The best performance from the three attempts was recorded for each test.

Squat Jump (SJ):Procedure:

Participants began in a position with 90-degree knee flexion and then jumped as high as possible.

Measurement:

The height of the jump, termed “flying height,” was measured using the Optojump system via infrared optical sensors (Globus; Microgate Ltd., Bolzano, Italy) as referenced by Bosco et al. [25].

Counter Movement Jump (CMJ):Procedure:

Starting from a standing position, participants were instructed to jump as high as possible.

Measurement:

As with SJ, the flying height was measured using the Optojump system via infrared optical sensors (Globus; Microgate Ltd., Bolzano, Italy), based on the methodology described by Bosco, Luhtanen, and Komi [25].

Five-Jump Test (FJT):Procedure:

This test consisted of five consecutive horizontal jumps. Participants started with their feet together, jumped forward while raising one knee and swinging the arms, then landed, and continued into the next jump. By the fifth jump, they would return to their starting position [26].

Measurement:

The total distance covered over the five jumps was measured.

Sprint Tests:Procedure:

From a standing start, participants ran two distances (10 m and 30 m) as quickly as possible in a straight line.

Measurement:

The time taken to complete the sprints was measured using photoelectric cells provided by Microgate Ltd. The first set of cells was at the start line, the second set was at 10 m, and the third set was at the 30 m finish line. Participants rested for three minutes between each attempt.

#### 2.3.4. Blood Sample Analysis

Before and after the 12-week WB-EMS intervention, fasting blood samples were drawn from the antecubital vein into heparinized tubes. The samples were then centrifuged at 2000 rpm for 25 min, after which the plasma was frozen at −40 °C pending analysis. Plasma CRP levels were determined using the immunoturbidimetric method. Levels of plasma lipids, such as total cholesterol, HDL-C, and triglycerides, were measured using enzymatic colorimetric methods. All of these parameters were analyzed using an Architect C8000 auto-analyzer and the associated reagent kits (provided by Abbott Laboratories, Abbott Park, IL, USA). The method of Friedewald et al. [27] was employed to calculate LDL-C.

#### 2.3.5. Sample Size

The sample size of our study was determined using the power analysis program G*Power 3.1 [28]. The priori power analysis was calculated using the F-test (ANOVA repeated measures [2 groups × 2 times]) and a related study that examined the effects of exercise training with WB-EMS [18]. Therefore, a total sample size of 30 participants was determined necessary to detect a power of 0.90, an alpha level of 0.05, and an effect size of 0.55.

#### 2.3.6. Randomization

The study sample was randomized using a computer random number generator, employing a simple randomization method to allocate participants into two distinct groups. The first group engaged in WB-EMS training (n = 16), while the second group was the control group (n = 15) (See Figure 1). We ensured that each participant had an equal chance of being assigned to either group, minimizing bias and enhancing the internal validity of the study. Randomized assignment was conducted after participants met the inclusion criteria.

### 2.4. Statistical Analysis

Data are presented as the mean ± standard deviation (SD). Normality and homogeneity of variances were checked by the Shapiro–Wilk and Levene tests, respectively. The reliability of all performance tests was assessed by the intraclass correlation coefficient and 95% confidence intervals during the familiarization sessions. A two-way analysis of variance (ANOVA) with repeated measures (2 groups × 2 times) was employed for all variables. Significant group × time interactions were followed by a post hoc Bonferroni test to identify significant pairwise comparisons. The effect sizes for time–group interaction effects were determined using partial eta squared (ηp^2^). Cohen’s d (d) was used to quantify the effect size between the pre-test and post-test means of each group or the post-test means between the two groups, categorizing differences as small (0.2), medium (0.5), or large (0.8) based on Cohen’s criteria [29]. All statistical analyses were conducted using SPSS software version SPSS 28.0 for Windows (SPSS, IBM Corporation, Chicago, IL, USA), with a significance level set at *p* ≤ 0.05 for all tests.

## 3. Results

### 3.1. Test Reliability

For all physical measures, test–retest reliability was determined during familiarization trials using pilot data from 28 participants collected on two different days. It is presented as the intraclass correlation coefficient and 95% confidence intervals in Table 3.

Prior to the intervention, there were no statistically significant differences between the WB-EMS and control groups for all variables under consideration, as presented in Table 4.

### 3.2. Physical Performance

Statistical analysis revealed a significant main effect of time for SJ, CMJ, FJT, and 30 m sprint time. Additionally, significant (time × group) interactions were observed for MAS, VO_2max_, SJ, CMJ, FJT, and 30 m sprint time (Table 4). Intragroup analysis of the WB-EMS group showed notable improvements in MAS (*p* = 0.010, ES = 0.49), VO_2max_ (*p* = 0.010, d = 0.48), SJ (*p* < 0.001, d = 1.15), CMJ (*p* < 0.001, d = 1.23), FJT (*p* < 0.001, d = 1.64), and 30 m sprint time (*p* = 0.001, d = 0.44) post intervention (Table 4). However, no significant changes were observed in the physical variables for the control group. When comparing the groups, the statistical analysis post training indicated that SJ (*p* = 0.017, d = 1.01), CMJ (*p* = 0.024, d = 0.95), and FJT (*p* < 0.001, d = 1.60) were superior in the training group compared to the control group (Table 4).

### 3.3. Cardiometabolic Markers

Our results revealed significant main effects of time and (time × group) interactions for all anthropometric and blood pressure variables (Table 4). After a 12-week WB-EMS training regimen, statistical analysis indicated notable improvements in the WB-EMS group for body mass (*p* = 0.001, d = 0.26), %BF (*p* = 0.002, d = 0.85), WC (*p* = 0.002, d = 0.36), SMM (*p* = 0.004, ES = 0.90) SBP (*p* = 0.003, d = 0.67), and DBP (*p* = 0.05, d = 0.46). These improvements were not observed in the control group (Table 4). Post intervention, the between-group comparison revealed that the WB-EMS group had a higher SMM (*p* = 0.020, d = 0.96) and lower SBP (*p* = 0.047, d = 0.82) compared to the control group (Table 4).

Significant main effects over time and (group × time) interactions were observed for total cholesterol (*p* = 0.033, ηp^2^ = 0.16 and *p* = 0.048, ηp^2^ = 0.14, respectively), triglycerides (*p* = 0.014, ηp^2^ = 0.21 and *p* = 0.049, ηp^2^ = 0.14, respectively), LDL-C (*p* = 0.033, ηp^2^ = 0.16 and *p* = 0.036, ηp^2^ = 0.16, respectively), HDL-C (*p* = 0.046, ηp^2^ = 0.14 and *p* = 0.049, ηp^2^ = 0.14, respectively), and C-reactive protein (*p* = 0.02, ηp^2^ = 0.20 and *p* = 0.049, ηp^2^ = 0.14, respectively).

In the intragroup analysis, there was a significant reduction in levels of total cholesterol (from 146 ± 20.0 to 138 ± 19.6 mg/dL, *p* = 0.004, d = 0.40), triglycerides (from 74.7 ± 26.9 to 64.5 ± 21.9 mg/dL, *p* = 0.002, d = 0.43), and LDL-C (from 87.3 ± 14.5 to 78.9 ± 17.4 mg/dL, *p* = 0.003, d = 0.54) after the WB-EMS training program (as shown in Figure 2). Conversely, HDL-C levels saw a significant increase (from 43.4 ± 6.47 to 46.1 ± 6.27 mg/dL, *p* = 0.005, d = 0.44). Additionally, the 12-week intervention resulted in a notable decrease in plasma CRP levels (from 2.07 ± 0.90 to 1.45 ± 0.45 mg/L, *p* = 0.002, d = 0.90) in the WB-EMS group (illustrated in Figure 3). In contrast, the control group did not exhibit any significant variations in these biochemical markers.

Following the training period, the between-group analysis highlighted that the WB-EMS group had significantly lower CRP levels (1.45 ± 0.45 vs. 2.10 ± 0.94 mg/L, *p* = 0.025, d = 0.94) compared to the control group (as depicted in Figure 3).

## 4. Discussion

To date, our study is the first to examine the effects of WB-EMS training on physical performance measures, as well as select cardiometabolic and inflammatory markers, in young females with obesity. Our primary findings indicate that a 12-week WB-EMS training program effectively enhances physical performance (including cardiorespiratory fitness, and jumping and sprinting capabilities), specific cardiometabolic parameters (such as blood pressure and lipid levels), and inflammatory markers (like CRP levels) in these individuals.

### 4.1. Physical Performance

Following the 12-week WB-EMS program, there was a notable increase in the cardiorespiratory fitness (VO_2max_) of obese females. Up to this point, no research has specifically explored the effects of this technology on young individuals with obesity. Existing studies have primarily focused on athletes and healthy adults [12,13,30]. For instance, a 10-week local electromyostimulation program targeting the quadriceps and hamstrings boosted VO_2max_ in healthy participants [30]. Interestingly, Amaro-Gahete, De-la, Sanchez-Delgado, Robles-Gonzalez, Jurado-Fasoli, Ruiz, and Gutierrez [12] observed an increase in VO_2max_ in runners after 6 weeks of WB-EMS, despite a marked reduction in their training volume. Even more compelling were the results from a longer 12-week training program [13].

In trained and highly fit individuals, WB-EMS has been linked to improvements in lower limb strength and enhancements in both jumping and running capabilities [8,12,13,31,32]. Our research is pioneering in demonstrating the positive impacts of a 12-week WB-EMS program on FJT performance, jumping height (for both SJ and CMJ), and 30 m sprinting time in obese young females. Furthermore, the WB-EMS group outperformed the control group (CG) in the FJT, SJ, and CMJ tests.

Consistent with our findings, prior WB-EMS studies spanning 6 to 14 weeks have reported enhancements in jumping and sprinting performances among trained participants [12,13,29,32]. The beneficial effects of WB-EMS on cardiorespiratory fitness and anaerobic performance can be attributed to several physiological adaptations. These include improved co-activation and coordination in the lower limbs (by leveraging various muscle groups, both agonists and antagonists) during exercises, and refined mechanical activities from enhanced motor unit recruitment and synchronization [12,13,29].

While our findings from the 10 m test remain somewhat elusive, the intricate technicalities of sprints, such as the maximum speed of cyclic motions, might offer some insight [32]. The lack of substantial progress in this specific test could be influenced by technical factors that impact the sprint’s duration. This is especially plausible since the 10 m test heavily relies on explosive strength and reactive speed.

### 4.2. Cardiometabolic Markers

Our study revealed that the WB-EMS program resulted in an increase in SMM and decreases in body mass, BMI, body fat, and waist circumference (WC) among young obese females. These results were consistent with previous studies emphasizing the effectiveness of the WB-EMS program in enhancing body composition indices and reducing obesity in obese individuals [11,33,34]. Specifically, Reljic, Konturek, Herrmann, Neurath, and Zopf [33] demonstrated the positive impact of a 12-week WB-EMS program on body mass and fat among adult women with obesity and metabolic syndrome. In a study of patients with obesity who had undergone bariatric surgery, Ricci, Di Thommazo-Luporini, Jürgensen, André, Haddad, Arena, and Borghi-Silva [34] reported similar benefits after just 6 weeks of WB-EMS. Importantly, these results were of clinical significance, as the reduction in BMI within the WB-EMS group corresponded to a two-degree decrease in obesity, surpassing the impact observed in the exercise-only group, which achieved a one-degree reduction in obesity.

Kemmler and colleagues found that elderly individuals with obesity, particularly postmenopausal women, experienced beneficial changes in body weight and overall abdominal fat after WB-EMS training [35,36,37]. A common trend across the majority of WB-EMS studies is a net weight reduction, attributed to a decrease in fat mass and an increase in SMM. A possible explanation for these effects is that a single WB-EMS session can elevate the resting metabolic rate, promoting fat metabolism for several hours afterwards, which contributes to significant body fat reduction [11].

Hortobágyi and Maffiuletti [38] posited that EMS protocols, lasting up to 6 weeks, can induce alterations in muscle metabolism. As the intensity of the impulses increased, more pronounced increases in SMM were observed [39]. Consistently, numerous studies have highlighted the positive impacts of WB-EMS on body composition when employed for durations of 12 weeks or more [39].

While individual responses may vary, the integration of electrical stimulation and exercise offers a comprehensive and potentially more effective strategy for addressing obesity [11,33,34]. It combines the benefits of muscle engagement, increased caloric expenditure, and metabolic improvements, making it a promising approach in the multifaceted battle against obesity [11,33,34].

In our study, the WB-EMS program led to a notable decrease in SBP, but there was no significant change observed in DBP. There is limited research on the effects of WB-EMS on cardiovascular indicators [34]. However, available studies have confirmed its positive impact on blood pressure in healthy young subjects [18,19]. The improvement in SBP might be attributed to significant beneficial changes in body composition and inflammatory markers [40]. Further studies are needed to elucidate the variations in blood pressure following a WB-EMS program.

Our study revealed a significant decrease in levels of total cholesterol, triglycerides, and LDL-C, coupled with an increase in HDL-C levels. A similar study by Reljic, Konturek, Herrmann, Neurath, and Zopf [33] found a noteworthy reduction in triglycerides following a 12-week WB-EMS program among obese women. In another study, a 10-week WB-EMS intervention showed positive effects on the lipid profiles of healthy sedentary women [41]. In contrast, Park, Na, Choi, Seon, and Do [18] reported no significant improvement in lipid profiles after a 6-week WB-EMS regimen in young women. These discrepancies might stem from differences in participant characteristics, such as gender, weight status, the duration and intensity of WB-EMS training, dietary habits, and other factors [42]. Our findings align with the hypothesis that combined electrical stimulation training enhances lipid oxidation, emphasizing triglycerides as a primary energy source for such training [41].

After a 12-week WB-EMS program, our research revealed a significant reduction in circulating CRP levels among young obese women. Currently, only two studies have investigated the effect of WB-EMS on CRP levels [17,43]. Both studies found no notable changes in CRP levels after a 12-week WB-EMS regimen in individuals with metabolic syndrome who were obese [43] and in advanced cancer patients [17]. The disparities between our findings and those of previous research could stem from various factors, such as gender, age, body weight status (whether normal, overweight, or obese), and physical capability. Notably, the earlier studies involved patient populations, whereas our study focused on young, otherwise healthy obese individuals. The decline in plasma CRP concentration in our study could be attributed to improvements in body fat and overall body mass.

However, it is important to recognize the limitations of our study. Firstly, we did not control for participants’ dietary habits, which might have influenced our outcomes. Moreover, while subjects underwent a pain threshold test to determine maximum WB-EMS intensity, pain is inherently subjective. As a result, we cannot definitively state that the current intensity used was sufficient to elicit the observed changes. Also, the suitability and comfort of the WB-EMS equipment for obese individuals may have varied. Ensuring that the equipment effectively accommodated participants of diverse body sizes and shapes was a real-time consideration. Finally, safety concerns specific to obese individuals, such as joint issues or musculoskeletal limitations, needed real-time attention. Adjustments in electrode placement and stimulation intensity were crucial to prevent discomfort or adverse effects.

The time efficiency, joint-friendly nature, and high degree of customization associated with this innovative training technology make it a potentially viable supplement to traditional exercise regimens. This is particularly relevant for individuals who may lack motivation or face challenges in participating in demanding resistance exercise protocols as part of their weight loss programs.

As we navigate the complexities of combating obesity and its associated challenges, WB-EMS emerges as a promising intervention that extends beyond the conventional boundaries of fitness training. Future research should continue to explore the mechanisms underlying these observed improvements and consider the long-term viability and effects of WB-EMS in obesity management. Research comparing WB-EMS with various physical activities or obesity management techniques is crucial to gauge its comparative benefits. Broadening the scope of research to encompass different ages, genders, and obesity severity levels is vital for a more complete understanding of WB-EMS’s effectiveness across a range of demographics. It is also important to evaluate the safety, practicality, and adherence of participants to WB-EMS in varied environments, such as clinical and home settings, to fully grasp its applicability in obesity treatment. Finally, future research should focus on the effects of WB-EMS on overall life quality, mental health, and emotional well-being, as these are essential aspects of comprehensive health care.

## 5. Conclusions

In conclusion, WB-EMS was proven to be an effective strategy for positively influencing body composition by increasing SMM and decreasing body mass, BMI, body fat, and waist circumference. This study also showed a positive effect on physical performance, including better cardiorespiratory fitness (MAS, VO_2max_) and especially better jumping (SJ, CMJ, FJT) and sprinting (30 m sprint) performances. The findings of the present study highlight the multifaceted benefits of WB-EMS training, particularly in the context of addressing obesity and ameliorating some selected cardiometabolic markers, such as total cholesterol, triglycerides, LDL-C and HDL-C, coupled with improved levels of inflammatory markers (CRP) in obese young females.

## Figures and Tables

**Figure 1 medicina-60-00230-f001:**
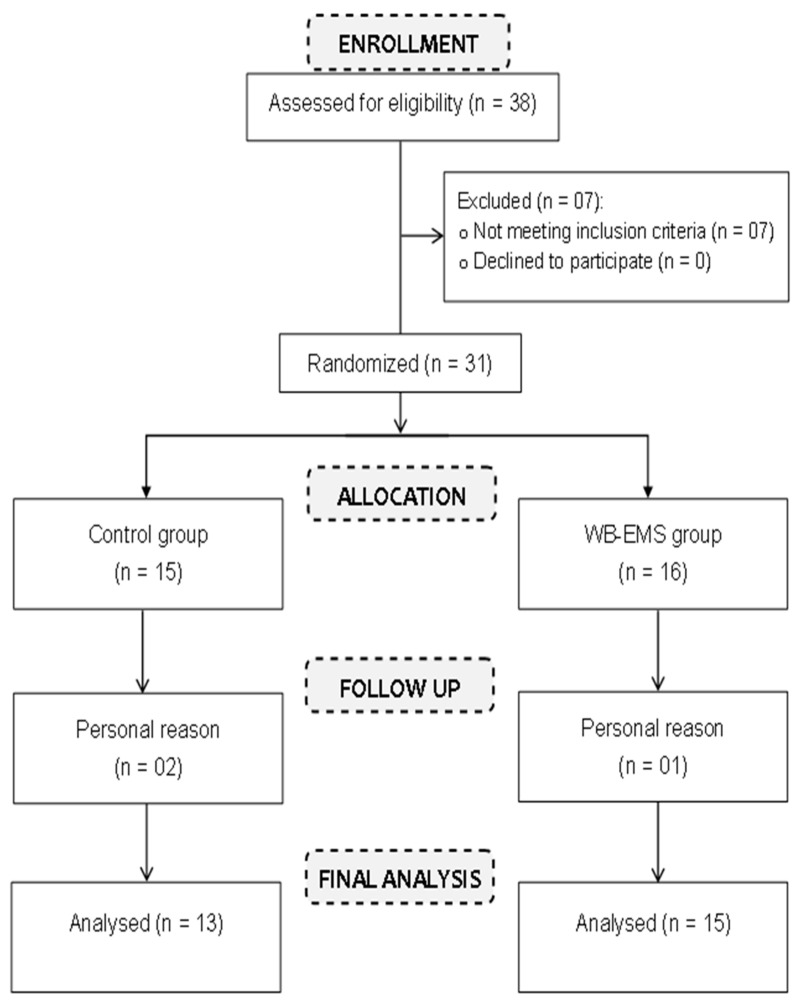
Study participants flow diagram.

**Figure 2 medicina-60-00230-f002:**
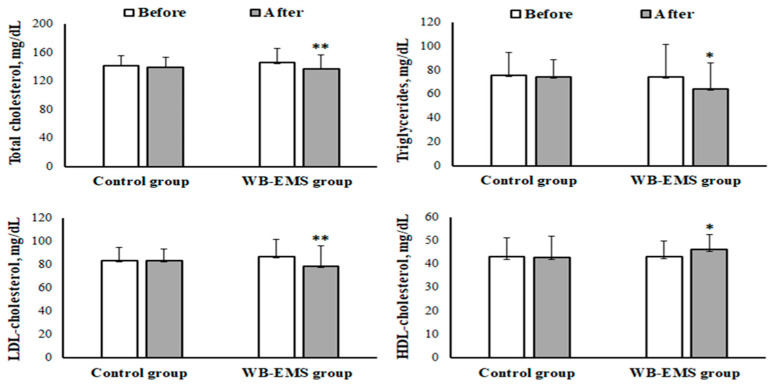
Mean ± SD of levels of total cholesterol, triglycerides, high-density lipoprotein cholesterol (HDL-cholesterol), and low-density lipoprotein cholesterol (LDL-cholesterol) before and after the whole-body electromyostimulation (WB-EMS) training program in obese young females. * *p* < 0.05; ** *p* < 0.01 (significant difference before and after program).

**Figure 3 medicina-60-00230-f003:**
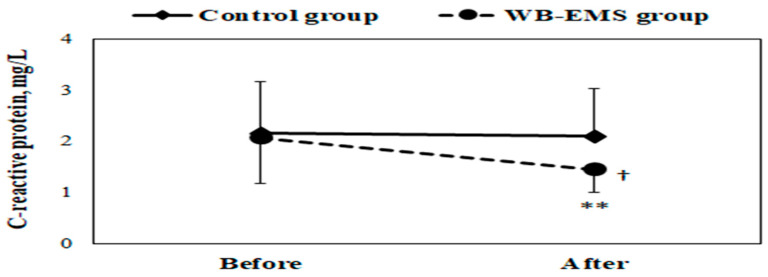
Mean ± SD of plasma C-reactive protein concentration in obese young females before and after the whole-body electromyostimulation (WB-EMS) training program. ^†^
*p* < 0.05 (significant difference between groups); ** *p* < 0.01 (significant difference before and after program).

**Table 1 medicina-60-00230-t001:** Mean ± SD of basal characteristics of whole-body electromyostimulation and control groups.

	WB-EMS Group (n = 15)	Control Group (n = 13)
Age (year)	18.6 ± 0.98	18.8 ± 0.83
Body mass (kg)	82.0 ± 10.6	82.1 ± 6.75
BMI (kg/m^2^)	31.3 ± 2.74	31.7 ± 2.90
Waist circumference (cm)	103 ± 8.60	105 ± 6.44
Body fat (%)	35.4 ± 2.16	35.8 ± 2.13
Skeletal muscle mass (kg)	31.1 ± 1.02	31.3 ± 0.98
SBP (mm Hg)	124 ± 3.96	124 ± 3.81
DBP (mm Hg)	77.1 ± 3.47	77.2 ± 3.72

BMI: body mass index; SBP: systolic blood pressure; DBP: diastolic blood pressure; WB-EMS: whole-body electromyostimulation.

**Table 2 medicina-60-00230-t002:** WB-EMS exercises.

Crunch	Bridge	Leg raise	Side plank
Back extension	Front plank	Lunge	Squat

**Table 3 medicina-60-00230-t003:** Reliability of performance tests.

	Intraclass Correlation Coefficient	95% Confidence Intervals
MAS (km/h)	0.777	0.573–0.890
VO_2max_ (mL/kg/min)	0.777	0.573–0.890
Squat jump (cm)	0.645	0.364–0.819
CMJ (cm)	0.766	0.555–0.885
Five-jump test (m)	0.638	0.354–0.814
10 m sprint test (s)	0.648	0.369–0.820
30 m sprint test (s)	0.925	0.846–0.965

CMJ, countermovement jump; MAS: maximal aerobic speed; VO_2max_: maximal oxygen consumption.

**Table 4 medicina-60-00230-t004:** Mean ± SD of anthropometric parameters and measures of physical performance before and following 12-week intervention in whole-body electromyostimulation and control groups.

	Control Group (n = 13)		WB-EMS Group (n = 15)		Time Effect			Group Effect			Interaction (Time × Group)		
	Before	After	Before	After	F	*p*	ηp^2^	F	*p*	ηp^2^	F	*p*	ηp^2^
Body mass (kg)	82.1 ± 6.75	81.3 ± 6.61	82.0 ± 10.6	79.4 ± 9.91 ***	17.8	<0.001	0.41	0.09	0.76	0.00	4.80	0.038	0.16
BMI (kg/m^2^)	31.7 ± 2.90	31.4 ± 2.84	31.3 ± 2.74	30.3 ± 2.60 ***	18.1	<0.001	0.41	0.58	0.46	0.02	4.76	0.038	0.16
Waist circumference (cm)	105 ± 6.44	106 ± 6.41	103 ± 8.60	100 ± 8.01 **	4.73	0.39	0.15	0.93	0.34	0.04	4.29	0.048	0.14
Body fat (%)	35.8 ± 2.13	35.2 ± 2.40	35.4 ± 2.16	33.5 ± 2.65 **	17.4	0.00	0.40	1.40	0.25	0.05	4.41	0.046	0.15
Skeletal muscle mass (kg)	31.3 ± 0.98	31.0 ± 0.75	31.1 ± 1.02	32.4 ± 1.95 **^,†^	5.24	0.03	0.17	2.213	0.15	0.08	9.95	0.004	0.28
SBP (mm Hg)	124 ± 3.81	124 ± 2.81	124 ± 3.96	122 ± 3.18 **^,†^	5.20	0.03	0.17	1.09	0.31	0.04	4.56	0.042	0.15
DBP (mm Hg)	77.2 ± 3.72	77.1 ± 2.47	77.1 ± 3.47	75.5 ± 3.68 *	4.53	0.04	0.15	0.48	0.49	0.02	3.73	0.060	0.13
Maximal heart rate (bpm)	197.8 ± 3.56	197.9 ± 2.76	198.0 ± 3.23	197.0 ± 3.14	1.37	0.25	0.05	0.07	0.79	0.00	1.86	0.180	0.07
MAS (km/h)	10.3 ± 1.17	10.2 ± 0.95	10.3 ± 1.03	10.8 ± 0.96 **	3.03	0.09	0.10	0.74	0.40	0.03	4.21	0.050	0.14
VO_2max_ (mL/kg/min)	37.7 ± 4.09	37.6 ± 3.32	37.9 ± 3.62	39.6 ± 3.35 **	3.00	0.09	0.10	0.74	0.40	0.03	4.22	0.050	0.14
Squat jump (cm)	14.7 ± 2.60	15.2 ± 2.79	14.9 ± 2.64	18.1 ± 3.12 ***^,†^	21.8	<0.001	0.46	2.41	0.13	0.09	11.3	0.002	0.30
CMJ (cm)	17.0 ± 3.47	17.3 ± 3.39	17.1 ± 2.53	19.8 ± 1.95 ***^,†^	22.9	<0.001	0.47	1.54	0.23	0.06	14.3	0.001	0.36
Five-jump test (m)	6.87 ± 0.62	6.94 ± 0.80	7.03 ± 0.64	8.03 ± 0.62 ***^,†††^	37.2	<0.001	0.50	6.95	0.014	0.21	28.1	0.000	0.52
10 m sprint test (s)	2.90 ± 0.42	2.95 ± 0.44	2.82 ± 0.36	2.70 ± 0.29	0.40	0.53	0.02	1.52	0.23	0.06	1.95	0.180	0.07
30 m sprint test (s)	6.77 ± 0.67	6.78 ± 0.73	6.72 ± 0.46	6.53 ± 0.43 **	5.39	0.03	0.17	0.49	0.49	0.02	6.46	0.017	0.20

BMI: body mass index; CMJ, countermovement jump; SBP: systolic blood pressure; DBP: diastolic blood pressure; MAS: maximal aerobic speed; VO_2max_: maximal oxygen consumption; WB-EMS: whole-body electromyostimulation. ^†^
*p* < 0.05; ^†††^
*p* < 0.001 (significant difference between groups); * *p* < 0.05; ** *p* < 0.01; *** *p* < 0.001 (significant difference before and after program).

## Data Availability

This article contains all of the information that was examined during this work.

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
