# Peer review of "The Effect of Whole-Body Electromyostimulation Program on Physical Performance and Selected Cardiometabolic Markers in Obese Young Females"

_medicina, 2024, doi:10.3390/medicina60020230_

Round 1
Reviewer 1 Report
Comments and Suggestions for Authors
Thank you for the opportunity to review this manuscript. It presents an interesting study about the impact of a Whole-body electromyostimulation training program on physical performance and cardiometabolic markers in young obese females.
The methodological approach is adequate.
The introduction and discussion present a clarifying revision of the literature. However, I have some comments and suggestions presented below.
General comments
Please, in the whole text change “whole-body electrostimulation” to “whole-body electromyostimulation”
P value in the whole manuscript must be expressed as p value.
Specific Comments
Title and abstract
Please change “Whole-body Electrostimulation” to “Whole-body electromyostimulation” and avoid the use of the acronym: “WB-EMS”
Line 26. Please, correct 27% to 28%, as it appears in the material and methods section.
Line 39. Conclusion in the abstract. Please correct according to the recommendation for the conclusions in the main text.
Introduction
Line 90. I think that some sentences must be added considering that WB-EMS should be treated as a supplemental training method, maybe of utmost importance in very deconditioned patients in the beginning. But I think that the use of the term alternative training method should be avoided, because I think that it is not probed that EMS could cover all the spectrum of benefits that the different modalities of physical activity have.
2. Materials and Methods
Line 118. Please change “individuals” to “women”.
2.2 Intervention. At the end of this subsection, please, include a sentence remembering that the control group made the same exercises as the other group but without the inclusion of the WB-EMS.
Line 154. “target eight muscle groups: the upper arms, chest, abdomen, latissimus, upper back, lower back, buttocks, and thighs” Please, clarify where the electrodes were applied. Upper arm is not a muscle group. I recommend for the future to try to imply a physiotherapist in the team that provide WB-EMS intervention. To help with the electrode’s application and the use of the EMS in a pathology such as obesity.
Table 2. Please avoid the use of the bold in the table.
2.3.1 Anthropometric assessment. Please, add some references to this paragraph supporting the measurement protocol of the height, the Tanita and the waist circumference.
2.3.5 Sample size. Please, add the reference of the previous manuscript that examined the effects of exercise training with WB-EMS that you have used for the calculation of the sample size. Line 241. Please specify that you used ANOVA, repeated measures, between factors for the calculation.
2.4 Statistical Analysis. Line 256. “Reliability of all variables”, please change to: “Reliability of all performance tests”. Please, specify what ICC did you use and the measures that you compared.
Line 261. For what comparison you calculated Cohen’s d?
4. Discussion
Line 393. Please avoid the discussion about HR in this line and the next lines. You have not presented data about HR in the results section.
Line 421. Please eliminate overweight.
5. Conclusion
Line 437. Please, clarify this conclusion, including the concrete variables that have shown inter-group differences after intervention: “influencing body composition physical performance (cardiorespiratory fitness and jumping and sprinting performance)”.
Line 441. From my point of view, the term alternative method should be avoided, it is better to use supplementary.
Lines 439-457. I think that this paragraph should be at the end of the discussion section as practical implications.
Line 472. Please remove: “This section is mandatory and should contain the main conclusions regarding the research.”
Author Response
Reviewer 1
- General comments
Please, in the whole text change “whole-body electrostimulation” to “whole-body electromyostimulation”.
P value in the whole manuscript must be expressed as p value.
Response: We express our gratitude to the reviewer for the suggestion. Subsequently, we have implemented the requested modifications throughout the entire text.
- Specific Comments
Title and abstract
Please change “Whole-body Electrostimulation” to “Whole-body electromyostimulation” and avoid the use of the acronym: “WB-EMS”.
Response: We thank the reviewer for the suggestion. Accordingly, the requested modifications are made.
Line 26. Please, correct 27% to 28%, as it appears in the material and methods section.
Response: We thank the reviewer for the suggestion. Accordingly, the requested modification is made.
Line 39. Conclusion in the abstract. Please correct according to the recommendation for the conclusions in the main text.
Response: We thank the reviewer for the suggestion. Accordingly, we made some modifications in conclusion in the abstract.
→Conclusion: A 12-week whole-body electromyostimulation program significantly improved body composition (skeletal muscle mass, body mass index, body fat, and waist circumference) physical performance (maximal oxygen consumption, jumping and sprint performance) and certain cardiometabolic (plasma level of lipids) and inflammatory markers (C-reactive protein) in obese young women. Further research is needed to explore the broader effects of whole-body electromyostimulation on physical and cardiometabolic health.
Introduction
Line 90. I think that some sentences must be added considering that WB-EMS should be treated as a supplemental training method, maybe of utmost importance in very deconditioned patients in the beginning. But I think that the use of the term alternative training method should be avoided, because I think that it is not probed that EMS could cover all the spectrum of benefits that the different modalities of physical activity have.
Response: We thank the reviewer for the suggestion. Accordingly, the term alternative is avoided, and we add some sentences.
→So, further studies are needed to confirm the role of this training method as a supplemental method that support and enhance traditional training programs in managing obesity and improving life quality for obese subjects. It could be served as a foundational step, helping deconditioned individuals gradually build strength, endurance, and overall fitness.
- Materials and Methods
Line 118. Please change “individuals” to “women”.
Response: We thank the reviewer for the suggestion. Accordingly, the requested modification is made.
2.2 Intervention. At the end of this subsection, please, include a sentence remembering that the control group made the same exercises as the other group but without the inclusion of the WB-EMS.
Response: We thank the reviewer for the suggestion. Accordingly, the sentence is added.
→The control group performed the same exercises as the WB-EMS group but without using electrical stimulation.
Line 154. “Target eight muscle groups: the upper arms, chest, abdomen, latissimus, upper back, lower back, buttocks, and thighs” Please, clarify where the electrodes were applied. Upper arm is not a muscle group. I recommend for the future to try to imply a physiotherapist in the team that provide WB-EMS intervention. To help with the electrode’s application and the use of the EMS in a pathology such as obesity.
Response: We thank the reviewer for the suggestion. We will follow your advice.
Table 2. Please avoid the use of the bold in the table.
Response: We thank the reviewer for the suggestion. Accordingly, the requested modification is made.
|
Crunch |
Bridge |
Leg raise |
Side plank |
|
Back extension |
Front plank |
Lunge |
Squat |
2.3.1 Anthropometric assessment. Please, add some references to this paragraph supporting the measurement protocol of the height, the Tanita and the waist circumference.
Response: We thank the reviewer for the suggestion. Accordingly, some references are added.
→Participants' body height was measured using an unstretched tape measure while they stood without shoes. Weight-related metrics, including body mass, BMI, body fat percentage (%BF), and skeletal muscle mass (SMM), were determined using a diagnostic scale (Tanita-DC-430U) [22]; during the test, individuals stand on the Tanita scale without shoes and light clothes, and the device measures the impedance of the electrical signal as it travels through the body. This information is then used to estimate the proportions of different body components. Waist circumference (WC) was measured using a non-deformable tape ruler, placed between the superior iliac crest and the lower rib margin [23].
2.3.5 Sample size. Please, add the reference of the previous manuscript that examined the effects of exercise training with WB-EMS that you have used for the calculation of the sample size.
Response: We added the requested reference, in the following sentence:
→ “2.3.5 Sample size
The sample size of our study was determined using the power analysis program G*Power 3.1 [28]. The priori power analysis was calculated using F-test (ANOVA repeated measures [2 groups × 2 times]) and a related study that examined the effects of exercise training with WB-EMS [19]. Therefore, a total sample size of 30 participants was determined necessary to detect a power of 0.90, an alpha level of 0.05, and an effect size of 0.55.” Please see in Sample Size section in the revised manuscript.
Line 241. Please specify that you used ANOVA, repeated measures, between factors for the calculation.
Response: We thank the reviewer for the suggestion. Accordingly, we added the following sentence:
→ “2.3.5 Sample size
The sample size of our study was determined using the power analysis program G*Power 3.1 [28]. The priori power analysis was calculated using F-test (ANOVA repeated measures [2 groups × 2 times]) and a related study that examined the effects of exercise training with WB-EMS [19]. Therefore, a total sample size of 30 participants was determined necessary to detect a power of 0.90, an alpha level of 0.05, and an effect size of 0.55.” Please see Sample Size section in the revised manuscript.
2.4 Statistical Analysis. Line 256. “Reliability of all variables”, please change to: “Reliability of all performance tests”. Please, specify what ICC did you use and the measures that you compared.
Response: We thank the reviewer for the suggestion. Accordingly, we added the following sentence: “Reliability of all performance tests was assessed by the intraclass correlation coefficient and 95% confidence intervals during the familiarization sessions.” Please see Statistical Analysis section in the revised manuscript.
Line 261. For what comparison you calculated Cohen’s d?
Response: We thank the reviewer for this comment. Accordingly, we added the following sentence: “Cohen’s d (d) was used to quantify the effect size, between the pre-test and post-test means of each group or the post-test means between the two groups, categorizing differences as small (0.2), medium (0.5), or large (0.8) based on Cohen's criteria [29].” Please see Statistical Analysis section in the revised manuscript.
- Discussion
Line 393. Please avoid the discussion about HR in this line and the next lines. You have not presented data about HR in the results section.
Response: We thank the reviewer for the suggestion. Accordingly, this part is delated from discussion.
Line 421. Please eliminate overweight.
Response: We thank the reviewer for the suggestion. Accordingly, we eliminate the term overweight.
- Conclusion
Line 437. Please, clarify this conclusion, including the concrete variables that have shown inter-group differences after intervention: “influencing body composition physical performance (cardiorespiratory fitness and jumping and sprinting performance)”.
Response: We thank the reviewer for the suggestion. Accordingly, we made some modifications in conclusion.
→ In conclusion, WB-EMS proves to be an effective strategy for positively influencing body composition by increasing SMM and decreasing body mass, BMI, body fat, and waist circumference. This study also shows a positive effect on physical performance; better cardiorespiratory fitness (MAS, VO2max) and especially better jumping (SJ, CMJ, FJT) and sprinting (30-m sprint) performances. The findings of the present study highlight the multifaceted benefits of WB-EMS training, particularly in the context of addressing obesity and ameliorating some selected cardiometabolic markers such as total cholesterol, triglycerides, LDL-C and HDL-C, coupled with improved level of inflammatory markers (CRP levels) in obese young females.
Line 441. From my point of view, the term alternative method should be avoided, it is better to use supplementary.
Response: We thank the reviewer for the suggestion. Accordingly, the term alternative is avoided and replaced by supplement.
→ The time efficiency, joint-friendly nature, and high degree of customization associated with this innovative training technology make it a potentially viable supplement to traditional exercise regimens.
Lines 439-457. I think that this paragraph should be at the end of the discussion section as practical implications.
Response: We thank the reviewer for the suggestion. Accordingly, we change the place of this paragraph we put this one in the end of discussion.
Line 472. Please remove: “This section is mandatory and should contain the main conclusions regarding the research.”
Response: We thank the reviewer for the suggestion. Accordingly, the sentence is removed.

Reviewer 2 Report
Comments and Suggestions for Authors
Your research is really interesting. I would like to ask you a few questions.
1. Is there a reason for not using the obesity index?
2. Please describe the exercise procedure in more detail. For example, including order and time.
3. An important factor in obesity is food. However, this study did not intervene on food intake. Of course, the limitations of the study were presented, but please more describe in the discussion the comparison with previous studies showing that electrical stimulation and exercise have an effect on reducing obesity.
Author Response
Reviewer 2
- Is there a reason for not using the obesity index?
Response: We thank the reviewer for the question which could give a better explanation in the text, accordingly some details about the BMI is added.
In our study, all participants have a BMI exceeding 30, categorizing them as obese according to the obesity index classification. Although this information was not initially included in the text, in response to your comment, we have incorporated the obesity index used for our participants into the text.
→Abstract: Methods: Twenty-eight obese females, aged over 18 years with BMI over 30
→ The inclusion criteria for participant eligibility encompassed women aged 18 years and above. Prospective participants were required to have no existing medical issues or be on medications that could significantly impact body fat or skeletal muscle mass. A BMI higher than 30 ……
- Please describe the exercise procedure in more detail. For example, including order and time.
Response: We thank the reviewer for the comment; accordingly, some details are added in the intervention part.
-The exercises following same order as presented below.
|
Crunch |
Bridge |
Leg raise |
Side plank |
|
Back extension |
Front plank |
Lunge |
Squat |
- Each participant trained in the same days and same time during the intervention. The training sessions was private for each woman.
- An important factor in obesity is food. However, this study did not intervene on food intake. Of course, the limitations of the study were presented, but please more describe in the discussion the comparison with previous studies showing that electrical stimulation and exercise have an effect on reducing obesity.
Response: We thank the reviewer for this comment. Accordingly, we added the following paragraph in the discussion.
→ While individual responses may differ, the integration of electrical stimulation and exercise presents a comprehensive and potentially more effective strategy for addressing obesity [34]. It combines the benefits of muscle engagement, elevated caloric expenditure and metabolic improvements making it a promising approach in the multifaceted battle against obesity [12,33,34].

Round 2
Reviewer 1 Report
Comments and Suggestions for Authors
I thank the authors for the implementation of the proposed corrections. I think you should still explain a little more where the electrodes were applied (line 159) to facilite the reproduction of the methods by other researchers.
Author Response
I thank the authors for the implementation of the proposed corrections. I think you should still explain a little more where the electrodes were applied (line 159) to facilite the reproduction of the methods by other researchers.
Response: We appreciate the reviewer's suggestion and have made modifications to the intervention accordingly.
2.2 Intervention
The experiment was overseen by licensed sports therapists with expertise in WB-EMS intervention. Each session commenced with a 5-minute warm-up on a bike ergometer at a moderate intensity. The WB-EMS program, set at 85 Hz with a duration of 350 μs, operat-ed intermittently, offering a 6-second impulse phase followed by a 4-second rest. The waveform was rectangular and delivered in a bipolar fashion. This program was de-signed to target eight muscle groups: the upper arms, chest, abdomen, latissimus, upper back, lower back, buttocks, and thighs. The stimulation is applied to the skin in proximity to the dermal tissue using wireless electrodes via Bluetooth technology [20]. Participants engaged in two 20-minute sessions per week for 12 weeks (24 sessions in total with at least two days’ rest between sessions).
